# Relationships between CD34-, CD105- and bcl-2-Expression Levels and Contrast-Enhanced Ultrasound-Based Differential Diagnosis of Adnexal Tumours

**DOI:** 10.3390/jcm12237372

**Published:** 2023-11-28

**Authors:** Marek Szymanski, Iwona Florczyk, Radoslaw Janicki, Piotr Bernard, Piotr Domaracki, Lukasz Brycht, Robert Szyca, Angelika Szymanska, Julia Paniutycz

**Affiliations:** 1Clinic of Gynaecology, Oncological Gynaecology and Gynaecological Endocrinology, 10th Military Research Hospital and Polyclinic, IPHC, Powstańców Warszawy Str. 5, 85-681 Bydgoszcz, Poland; m.szymanski1313@gmail.com (M.S.);; 2Department of Obstetrics, Female Pathology and Oncological Gynaecology, University Hospital No. 2, Collegium Medicum in Bydgoszcz, Nicolaus Copernicus, University in Toruń, Ujejskiego St. 75, 85-168 Bydgoszcz, Poland; 3NZOZ Centrum Medyczne „Genesis”-Clinic of Infertility Treatment, Waleniowa Str., 24, 85-435 Bydgoszcz, Poland; 4Department of Internal Medicine, Independent Public Health Care, Tadeusza Kościuszki 6, 88-300 Mogilno, Poland; 5Individual Medical Practice Iwona Florczyk, Wesolowskich 5/3, 88-300 Mogilno, Poland; 6Department of General and Oncological Surgery, Specialist Hospital in Włocławek, 49 Wieniecka Street, 87-800 Włocławek, Poland; 7Clinic of Surgery and Oncological Surgery, 10th Military Research Hospital and Polyclinic, IPHC, 85-681 Bydgoszcz, Poland; 8Faculty of Medicine, Medical University in Poznań, 61-701 Poznań, Poland; 9Gynecology, Obstetrics and Pregnancy Pathology Department, Multi-Speciality City Hospital in Bydgoszcz, 19 Szpitalna Street, 85-172 Bydgoszcz, Poland

**Keywords:** ultrasonography, ovarian cancer, contrast-enhanced ultrasound, adnexal tumors, CEUS, CD34, CD105, bcl-2

## Abstract

The relationships between CEUS parameters of adnexal tumours and postoperative immunohistochemical assessments of CD34, CD105 and bcl-2 were analysed. This study aimed to investigate whether contrast-enhanced ultrasonography (CEUS) parameters depend on the microvascular density of the tumour lesion found after surgery. Fifty-one patients with a diagnosis of adnexal tumours were included in this single-centre, prospective study. Participants underwent preoperative CEUS (contrast-enhanced ultrasound). Colour Doppler enhancement characterisation parameters (Ystart, Ymax and S) were determined. Immunohistochemical examination of histological specimens of the adnexal lesions was then carried out to determine the expression levels of the CD34, CD105 and bcl-2 proteins. Relationships between the aforementioned parameters were investigated. No significant statistical correlations were observed between CD34, CD105 and bcl2 expression levels and CEUS parameters, independently of whether the operated lesion was malignant or benign. Transvaginal CEUS is diagnostic for the detection of pathological neoplastic vascularisation of an adnexal lesion independent of the density of microcapillaries found postoperatively.

## 1. Introduction

According to the latest report of the International Agency for Research on Cancer, ovarian cancer is the fourth most common gynaecological cancer, characterised by a high mortality rate [1,2]. Most patients with ovarian cancer are diagnosed at an advanced stage, when curative therapy is no longer effective [3]. Almost 10% of women will undergo exploratory surgery during their lifetime to assess ovarian masses [4]. Prompt diagnosis of malignant ovarian tumours and referral to a gynaecological oncologist can prolong patient survival. However, no single standardised method is available to diagnose ovarian malignancy [5,6]. Previous studies have used biochemical and immunohistochemical markers and imaging modalities to diagnose ovarian cancers. In particular, transvaginal contrast-enhanced ultrasonography (CEUS) has demonstrated promising results, enabling preoperative detection of ovarian tumours with a high degree of accuracy [7].

Angiogenesis is characterised by the formation of new blood vessels in healing and cancerous tissues, which is essential for regenerative growth, survival of cancer cells, and metastasis. CD34 expression, which indicates microvessel density, is significantly higher in malignant lesions than in normal tissues. Furthermore, CD105, also known as endoglin, exhibits remarkably high specificity for newly developing neoplastic vessels, making it a useful indicator of angiogenesis and breast cancer metastasis [8]. The endoglin gene is usually expressed at low levels in resting endothelial cells. However, its expression increases following the initiation of neoangiogenesis and activation of endothelial cells, as observed within tumour vessels [9]. The ability of cancer cells to stay alive depends on their ability to evade programmed cell death, or apoptosis. Among apoptotic pathways, the intrinsic pathway involves the disruption of mitochondrial function. The main participants in this pathway are members of the B cell CLL/lymphoma 2 (BCL2) family [10].

As mentioned above, the concentration of CD34, CD105 and bcl-2 may indicate the micro vascularization of an adnexal tumour. CEUS is an examination that improves the visualisation and the number of vessels that can be observed, but especially allows these enhancement changes to be observed during the examination. The pathological structure of the tumour vessels (e.g., different arterial wall structure, venous sinuses, arteriovenous junctions, etc.) has an impact on contrast washout time. Thus, we are able to detect this preoperatively and attempt to differentiate tumour lesions. The present study investigated relationships between contrast kinetics in pathological tumour vessels in CEUS during preoperative examination and postoperative expression levels of CD105, CD34, and bcl-2 in patients with ovarian cancer.

## 2. Materials and Methods

### 2.1. Patients

A single-centre study enrolled 51 patients (aged between 25–72 years) with unclear diagnosis of an ovarian tumour, referred for elective surgery for unilateral or bilateral adnexal masses. The patients underwent preoperative computer-assisted examination via CEUS, among whom 33 were postoperatively diagnosed with malignant tumours and 18 with benign tumours. In each patient, postoperative immunohistochemical assessment of CD34, CD105 and bcl-2 was performed. Written informed consent was acquired from all patients before enrolment.

### 2.2. CEUS

Patients diagnosed with adnexal tumours were referred to the clinic. Diagnostics based on the International Ovarian Tumour Analysis (IOTA) criteria did not clearly determine whether the tumours were benign or malignant. During hospitalization, each patient further underwent CEUS examination. If the patient was diagnosed with more than one lesion, only the lesion with the highest vascular density was submitted to the analysis.

CEUS (128 XP/4; Acuson, Acuson Corporation, Mountain View, CA, USA; Voluson E6; General Electric Company, Boston, MA, USA) was performed using an endovaginal probe at a frequency of 7.5 MHz. The contrast agents Levovist (400 mg/mL) and SonoVue (8 μL/mL) were prepared according to the manufacturers’ instructions and administered as bolus doses over 10 s. CEUS examination was performed by a cross-section of a selected solid part of the tumour. An aim was made to maintain this cross-section through the whole examination. Characteristic areas of the tumour cross-section serving as landmarks were used to assist in this. All examinations were performed by a single physician (M.S.).

The ultrasound images were analysed using computer software (CQ program; Kinetic Imaging, Liverpool, UK), and the obtained parameters were statistically analysed. The fractional colour Doppler area (%CDA) was determined from the images. Appropriately acquired images were analysed to determine the %CDA in three successive phases: an initial value following contrast administration, a rapid increase to a maximum value, and a slower decrease toward the initial value.

The curve obtained from computerised analysis of CEUS images was used to determine the following values: initial %CDA (Y_start_), %CDA at the time of contrast administration (T_kts_), coordinates of the maximum growth point (T_max_ and Y_max_), relative increase in %CDA from the time of contrast administration to the time of the maximum %CDA value (δ = Y_max_/Y_start_), drop duration (T_sp_ = T_end_ − T_max_), falling speed (D = tgϕ, where tgϕ = (Y_end_ − Y_max_)/T_sp_) and final %CDA (T_end_).

Except for incorrectly acquired curves, all curves obtained from patients with malignant tumours were similar in shape to the curve presented in Figure 1. The initial (Y_start_) and maximum (Y_max_) values obtained from the curves were compared between patients with malignant tumours (Figure 2, Examples A and B) and those with benign tumours (Figure 2, Examples C and D).

The S parameter was used to evaluate the luminous intensity of the images, as in the following integral:(1)$$S=\int_{0}^{T*}F_{\%CAD}\;(t)dt,$$
where T* is the upper integration limit, which is common among all analysed graphs. S is the most appropriate parameter for comparison, as it represents a collective evaluation and is not significantly influenced by individual F_%CDA_ values.

### 2.3. Immunohistochemistry

The patients underwent surgery within 2 days of undergoing CEUS. The type of surgery was decided on the basis of the intraoperative histopathological diagnosis, patient age, clinical stage of disease, and general condition of the patient. The patients underwent surgery ranging from removal of the lesion alone to extended hysterectomy with pelvic lymph node dissection and removal of the greater omentum and appendix, often accompanied by segmental resection of the intestines and stoma creation.

The ovarian tumours obtained during surgery were subjected to immunohistochemical analysis to determine vessel density using specific antibodies. The tumour specimens were incubated with monoclonal mouse primary antibodies directed against human antigens: CD34 (1:25; class II; clone QBEnd 10; DakoCytomation, Glostrup, Denmark), endoglin/CD105 (1:50; clone 4G11; Novocastra, Newcastle upon Tyne, UK), and bcl-2 (1:80; bcl-2/100/D5 clone; Novocastra). Angiogenesis was assessed based on the expression levels of antigens CD34 and CD105 and the level of the apoptosis modulator bcl2.

The relationships between the expression levels of CD34, CD105 and bcl-2 and CEUS parameters (Y_start_, Y_max_, and S) obtained from the computer software were analysed.

## 3. Results

### 3.1. S Parameter

The means and standard deviations of the Y_start_, Y_max_ and S parameters are shown in Table 1. The parametric Cochran–Cox test was used to compare means across groups with different variances. For the Y_start_ and S parameters, a non-parametric Mann–Whitney U test was used for group comparisons.

The Cochran–Cox test revealed that the mean Y_max_ was significantly higher in patients with malignant tumours than in those with benign tumours. The Mann–Whitney U test also showed that Y_start_ and S differed significantly between these groups. Furthermore, the mean values were significantly higher for malignant tumours than for benign tumours. Therefore, we hypothesised that there might be significant correlations among these parameters. To verify this hypothesis, we performed correlation analysis for the entire study cohort (n = 51) presented in Figure 3.

We detected strong positive correlations among the parameters obtained from the computerised image analysis. Therefore, we evaluated the results of the computerised image analysis using the integral S parameter. The S parameter was <90 for benign tumours, whereas higher values were observed in malignant neoplasms (Figure 4). Conversely, for S parameter values of 0–30, malignant tumours were less common than benign tumours.

### 3.2. Correlation of the S Parameter with CD34-, CD105- and BCL2-Expression Levels

The results of correlation analyses between S parameter values obtained from computerised image analysis and CD34- or CD105-expression levels are shown in Figure 5.

No relationships were found between the S parameter and antigens CD34 and CD105. To explore a possible association between bcl-2 levels and the S parameter, we performed correlation analysis and an independent chi-square test. The S parameter was categorised according to its median value. Significance was evaluated at a level of *p* < 0.05, and the degree of freedom (df = 3) was 7.81. No correlation was detected between the S parameter and bcl-2 expression presented in Table 2.

These results imply that there were no significant relationships between the CD34, CD105 or bcl-2 levels and contrast-flow parameters obtained from computerised image analysis in this study. Antigen expression levels are independent prognostic factors of vessel density in neoplastic lesions. For S parameter values ≤ 30, benign tumours were threefold more common than malignant tumours. Similarly, S parameter values > 90 were associated with a high probability of a malignant tumour.

## 4. Discussion

Identification of ovarian lesions is essential in gynaecological practice, including distinguishing between benign and malignant adnexal tumours and, if necessary, selecting the appropriate surgical treatment. Almost 2% of adnexal masses are carcinomas or borderline tumours [11]. Correct diagnosis of pathological adnexal masses is essential for appropriate treatment planning. Several imaging techniques can be used to evaluate adnexal lesions before treatment, including transvaginal ultrasonography, magnetic resonance imaging, computed tomography, and positron emission tomography combined with computed tomography. Transvaginal ultrasonography is considered the gold standard diagnostic technique for early assessment of adnexal lesions in the pelvis minor [12,13].

International guidelines recommend ultrasound as the first-line technique for the diagnosis of patients with a suspected isolated ovarian mass [14]. However, conventional power Doppler ultrasonography has low sensitivity (50–80%) and specificity (80–90%) for distinguishing between benign and malignant tumours [15,16,17]. Ovarian cancer is associated with an unfavourable prognosis, with 5-year survival rates ranging from 27% to 16% for FIGO stages III and IV, respectively. This poor prognosis is mainly attributable to delayed diagnosis [18]. Therefore, early diagnosis is essential to improve the prognosis and prolong the life expectancy of patients. Multiple studies have evaluated the role of CEUS in the diagnosis of ovarian tumours, and found that ultrasound parameters are correlated with tumour angiogenesis and tumour progression [19,20,21,22]. CEUS analyses tissue perfusion, which provides important information related to angiogenesis. Angiogenesis, a major step in the proliferation of ovarian cancer cells, is characterised by physical changes that are visualised in ultrasound images. Some studies have suggested that the quantitative parameters of CEUS (time-curve intensity) can distinguish between incipient and malignant adnexal lesions [22]. Fleischer et al. [23] and Testa et al. [24] found that the peak intensity and area under the curve were significantly higher for malignant tumours than for borderline or benign tumours. Sconfienza et al. [25] demonstrated that malignant changes in the uterine adnexa were characterised by a significantly shorter time to peak compared to benign lesions. Furthermore, Orden et al. [20] and Marret et al. [21] showed that the area under the curve was highest for malignant invasive tumours, although the time to peak showed no consistent associations [26].

Several studies have evaluated biochemical markers of adnexal tumours. Ping et al. [27] demonstrated that CD105-expressing tumours were more likely to be at a higher stage (*p* = 0.02). Similar to our results, Szubert et al. [28] found that CD105 had no significant predictive value. Taskiran et al. [29] assessed the prognostic value of endoglin (CD105) in ovarian cancer, and found that endoglin levels were significantly related to advanced and suboptimal cytoreduction (*p* = 0.02 and 0.05, respectively). In this study, endoglin was an independent predictor of poor survival, implying that it may be used as an anti-angiogenic therapy.

A previous study hypothesised that the bcl-2 family could be targeted for individualised treatment of ovarian cancer [30]. Munjishvili et al. [31] suggested that several immunohistochemical markers, including bcl-2 and CA-125, are useful for differential diagnosis of borderline ovarian tumours with low-grade serous carcinomas and benign cystadenomas. Drugs that target the anti-apoptotic bcl-2 protein have also proven effective. Over the past few decades, the identification of various small-molecule inhibitors of bcl-2 anti-apoptotic proteins has contributed to their clinical applications [32]. Urinary bcl-2 levels are elevated in patients with ovarian cancer and have diagnostic and prognostic significance. Further research on urinary bcl-2 as a biomarker for ovarian cancer, either alone or in combination with other markers, is warranted [33]. Previous studies have also suggested that anti-ovarian autoantibodies provide important diagnostic information and are correlated with the aggressiveness of ovarian cancer [34].

Zahn et al. [35] demonstrated that serum CA19-9, CA-125 and CEUS levels were significantly higher, and the time to peak on CEUS was significantly shorter in patients with malignant ovarian epithelial tumours than in those with serous ovarian cancer. Serum CA19-9 and CA125 levels alone cannot distinguish between benign and malignant tumours, as levels can be elevated in several other conditions including endometriosis, tubo-ovarian abscesses and fibromas [36]. The combination of CA-125 levels and CEUS is widely used for preoperative diagnosis of ovarian tumours, as CEUS alone has lower diagnostic accuracy for distinguishing between benign and malignant adnexal lesions. However, CEUS is useful for the diagnosis of ovarian cancer [37,38]. The contrast washout patterns on CEUS can also distinguish between benign and malignant tumours [39], and real-time CEUS can be used to visualise the microvascular component of adnexal masses and provide improved blood flow reflection [40]. Although CEUS can distinguish between benign and malignant adnexal lesions, there is significant overlap between benign and borderline tumours. However, CEUS can distinguish between invasive malignancies and other lesions [25].

CA-125 is the most widely used biochemical marker. In combination with CEUS, it can identify ovarian lesions with high accuracy. Future studies should evaluate the usefulness of combinations of several independent biochemical, immunohistochemical and imaging factors for the diagnosis of adnexal lesions.

Tumour markers and adnexal changes on CEUS provide important diagnostic and predictive information that can be used to identify ovarian lesions with high accuracy. Nevertheless, complete agreement between pathological assessment and imaging studies is not possible, given that none of the investigations has perfect accuracy. Although the final diagnosis of ovarian masses is based on histopathological examination, preoperative differentiation between benign and malignant lesions is essential for surgical planning [26].

## 5. Conclusions

The contrast washout time on CEUS is not associated with the levels of the markers CD34, CD105 and bcl2; therefore, CEUS and immunohistochemical analysis provide independent information. CEUS parameters, in particular the washout time, are independent of the vascular density of adnexal tumours determined postoperatively in either malignant or non-malignant lesions. CEUS examination allows preoperative assessment of the pathological vascularisation of the lesion and is thus useful in their differentiation in difficult clinical cases. Further investigation is needed in order to examine more research and to confirm these findings. Another issue related to this study would be discussion on the idea of extending the IOTA criteria to include CEUS.

## Figures and Tables

**Figure 1 jcm-12-07372-f001:**
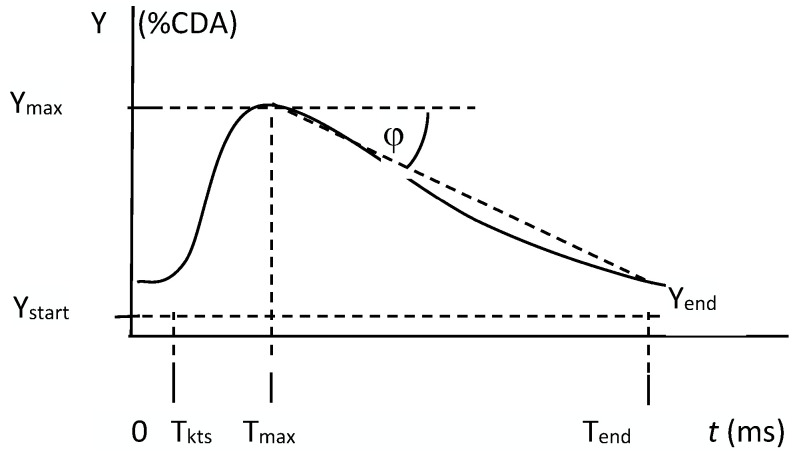
Curve obtained from computerised analysis of contrast-enhanced ultrasound (CEUS) images.

**Figure 2 jcm-12-07372-f002:**
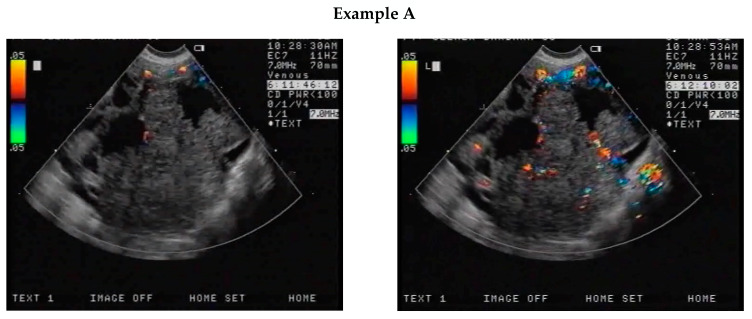
Examples of images of CEUS with contrast agent before contrast was applied and an image of maximum enhancement. The shape of the curve underneath shows the variations over time of the %CDA examination. Examples A and B show results of examination of malignant lesions. Examples C and D show results of examinations of benign lesions.

**Figure 3 jcm-12-07372-f003:**
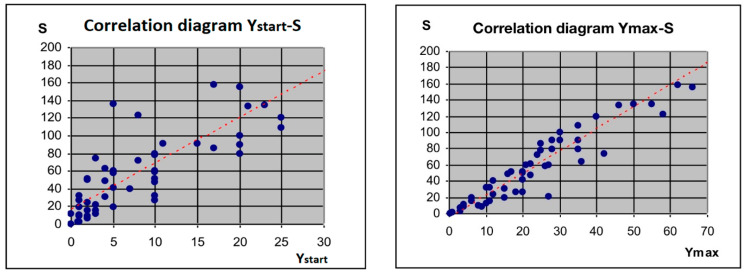
Correlation analysis of parameters obtained from the computerised image analysis.

**Figure 4 jcm-12-07372-f004:**
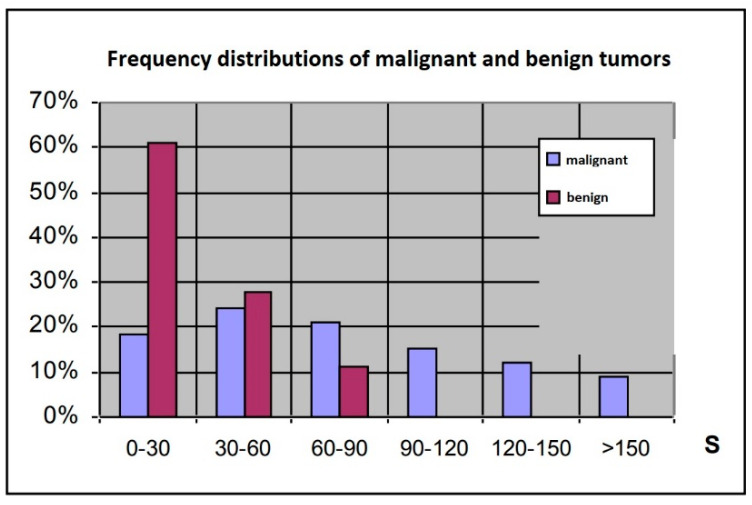
Frequency distributions of the S parameter for patients with malignant and benign tumours.

**Figure 5 jcm-12-07372-f005:**
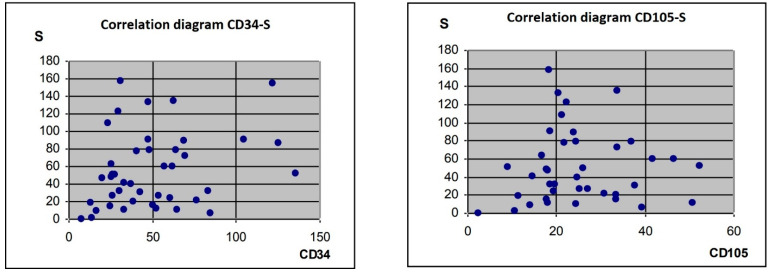
Correlation analyses of S parameter values and of CD34 or CD105 expression.

**Table 1 jcm-12-07372-t001:** Means and standard deviation (SD) of the Y_start_, Y_max_ and S parameters.

		Y_start_	Y_max_	S × 10^−3^
Malignant tumours (n = 33)	Mean	10.91	31.30	80.5
	*SD*	8.39	16.91	50.6
	Coefficient of variation	76.9%	54.0%	62.9%
Benign tumours (n = 18)	Mean	4.27	11.29	25.00
	*SD*	5.30	8.78	22.67
	Coefficient of variation	124.1%	77.8%	90.7%
Cochran–Cox test C_kr_ = 2.06	*C*	–	5.56	–
	*p*	–	<0.0001	–
Mann–Whitney U test U_kr_ = 1.96	*U*	3.47	–	4.26
	*p*	0.0005	–	<0.0001

**Table 2 jcm-12-07372-t002:** Results of correlation analysis between the S parameter and bcl-2 expression.

		S (Me = 46.9)		
		£ Me	>Me	Sum
bcl-2-intens	0	7	10	17
	1	5	6	11
	2	6	2	8
	3	2	3	5
Sum		20	21	41
Chi-square test		c^2^ = 2.80		
		*P* = 0.42		

## Data Availability

Data available on request due to restrictions eg privacy or ethical.

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
