# Peer review of "Relationships between CD34-, CD105- and bcl-2-Expression Levels and Contrast-Enhanced Ultrasound-Based Differential Diagnosis of Adnexal Tumours"

_jcm, 2023, doi:10.3390/jcm12237372_

Round 1

Reviewer 1 Report

Comments and Suggestions for Authors

There is an interesting idea to find a correlation between IHC markers expression and USD-based diagnosis. Although the authors did nor revealed any reflection of these markers levels during preoperative USD investigations, this m=negative result also contribute into the knowledge of the USD potential power in differential diagnosis. At the same time these markers do not play a significant role in histology-based diagnoses for the most widely spread adnexal tumors. So, it is not surprisingly that none of the correlation was revealed. It would be better to use at least proliferative markers to compare or some histologic criteria which are used to applying for differential diagnosis for benign cystadenomas, borderline and malignant tumors. In addition, the authors did not provide a relevant description of the IHC-based expression assessment.

Author Response

Dear Reviwer,

I am really greatful for your contribution and constructive comment to our manuscript.  Thank You for pointing some issues. Please find the detailed responses below and the corresponding corrections highlighted in the re-submitted files which were corrected accordingly to reviwers comments.

CommentAt the same time these markers do not play a significant role in histology-based diagnoses for the most widely spread adnexal tumors.

This immunohistochemical markers were proposed by the pathology department with which we cooperated. As well, we found these markers in other reaserch studies assesing adnexal tumors. That is why we decied to investigate CD34, CD105 and bcl-2, which determine vessel density.

CommentIt would be better to use at least proliferative markers to compare or some histologic criteria which are used to applying for differential diagnosis for benign cystadenomas, borderline and malignant tumors

Enrolling patients to our study was dictated by the difficulty in diagnosing adnexal tumors. Out of approximately 120 patients, only patients with an unclear diagnosis were included in the study, where it was not possible to determine whether the tumor was malignant or benign based on the IOTA criteria. In our study, we aimed to present the usefulness of CEUS as examination in diagnosing difficult cases of adnexal lesions and simultaneously the relationship with immunohistochemical markers describing vascular density.

I hope my response was useful and clarifying.  We strongly believe that this manuscript is suitable for publishing in the Jorunal of Clinical Medicine. 

Reviewer 2 Report

Comments and Suggestions for Authors

The present study investigated relationships between contrast kinetics in pathological tumour vessels on CEUS and expression levels of CD105, CD34, and bcl-2 in patients 74 with ovarian cancer. Overall, this study is well designed and have useful information. However, some reinfinements are needed.

 1- introduction should be improved by clear mention of your hypothesis and research gap.

2- methodology and results section should have illustrative images of contrast enhanced ultrasonography of both types of tumors as well as immunohistochemical findings.

3-please provide a full-detailed illustration of the Doppler investigation procedures to help the readers.

4- On what basis did you consider the sample size of the study was enough for the soundness of the study.

5- please present the last paragraph of the discussion section separately as a conclusion section

Author Response

Dear Reviwer,

thank you very much for taking the time to review this manuscript. Please find the detailed responses below and the corresponding corrections highlighted in the re-submitted files. 

Comment 1Introduction should be improved by clear mention of your hypothesis and research gap.

According to Your suggestion I modified the introduction in order to clarification.

Comment 2 and 32- methodology and results section should have illustrative images of contrast enhanced ultrasonography of both types of tumors as well as immunohistochemical findings.

3-please provide a full-detailed illustration of the Doppler investigation procedures to help the readers.

According to Your suggestion I have attached the illustrations to the manuscript. Unfortunately, we have no illustration of the immunohistochemical findings.

Comment 4On what basis did you consider the sample size of the study was enough for the soundness of the study.

The number of patients was dictated by the difficulty in diagnosing adnexal tumors. We initially screened approximately 120 patients, but qualified those with significant variation in lesions on usg images. The aim of the study was to investigate whether CEUS parameters depend on the density of microvessels found in postoperative specimens and independently of if the lesion on histopathological examination was malignant or benign.

Comment 5 - „please present the last paragraph of the discussion section separately as a conclusion section”

I have done accordingly to Your suggestion.

I hope my response was useful and clarifying.  We strongly believe that this manuscript is suitable for publishing in the Jorunal of Clinical Medicine.

Round 2

Reviewer 1 Report

Comments and Suggestions for Authors

All the corrections were completed.